# Ultralong phosphorescence cellulose with excellent anti-bacterial, water-resistant and ease-to-process performance

Xin Zhang[1,2], Yaohui Cheng[1,2], Jingxuan You[1,2], Jinming Zhang [1✉], Chunchun Yin[1,2] & Jun Zhang [1,2✉]

Herein, we present a phosphorescent cationized cellulose derivative by simply introducing ionic structures, including cyanomethylimidazolium cations and chloride anions, into cellulose chains. The imidazolium cations with the cyano group and nitrogen element promote inter-system crossing. The cyano-containing cations, chloride anions and hydroxyl groups of cellulose form multiple hydrogen bonding interactions and electrostatic attraction interactions, effectively inhibiting the non-radiative transitions. The resultant cellulose-based RTP material is easily processed into phosphorescent films, fibers, coatings and patterns by using eco-friendly aqueous solution processing strategies. Furthermore, after we construct a cross-linking structure by adding a small amount of glutaraldehyde as the cross-linking agent, the as-fabricated phosphorescent patterns exhibit excellent antibacterial properties and water resistance. Therefore, considering the outstanding biodegradability and sustainability of cellulose materials, cellulose-based easy-to-process RTP materials can act as antibacterial, water-resistant, and eco-friendly phosphorescent patterns, coatings and bulk materials, which have enormous potential in advanced anti-counterfeiting, information encryption, disposable smart labels, etc.

[1] CAS Key Laboratory of Engineering Plastics, CAS Research/Education Center for Excellence in Molecular Sciences, Institute of Chemistry, Chinese Academy of Sciences (CAS), 100190 Beijing, China. [2] University of Chinese Academy of Sciences, 100049 Beijing, China. ✉email: zhjm@iccas.ac.cn; jzhang@iccas.ac.cn

Phosphorescent materials have many advantages, such as a long emission life, a high signal-to-noise ratio, no background fluorescence and scattered light interference, and a large Stokes shift. Thus, they exhibit huge potential in biological imaging, information encryption, anticounterfeiting, light-emitting diode (LED) lighting, and lasers[1–4]. Compared with traditional metal-based phosphorescent materials, organic room-temperature phosphorescence (RTP) materials have low toxicity, low cost, and high flexibility. As a result, they have garnered tremendous interest in biomedical applications, information security, anticounterfeiting, flexible displays, comfortable wearable devices, photochemical catalysis, etc[5–19].

Two essential issues must be solved to achieve phosphorescence. The first is to improve the intersystem crossing (ISC)[20]. Because the singlet excited state and triplet excited state have different electron spin directions, the ISC process is forbidden. By introducing heavy atoms[21–26], the spin–orbit coupling (SOC) effect can be enhanced to promote the ISC. The second issue is to suppress the non-radiative transitions[27,28]. The triplet excited state can release energy in the form of non-radiative transitions caused by the motions of molecules or groups, e.g., rotation, vibration and collision, or forming interactions with the external environment, such as oxygen and solvents, which results in phosphorescence quenching. To achieve phosphorescence emission, organic materials usually must form crystal structures[29–37], supramolecular assemblies[38,39], encapsulation structures[40,41], or crosslinked networks[42] to confine the motions of molecules/groups and isolate oxygen. However, due to the formation of a rigid or complex structure, the obtained RTP materials generally exhibit poor processability; therefore, it is difficult to satisfy various practical requirements. Recently, Ma et al. synthesized a series of printable phosphorescent polymer materials[43–45]. An et al. prepared water-soluble phosphorescent polymer materials[46], which provided opportunities for their processing, forming, and application. However, synthetic polymers are generally difficult to realize a complete biodegradation, which has recently been an essential feature of new materials[47–50]. Moreover, the usage quantity of phosphorescent materials as labels and coatings is too small to recycle or reuse. Therefore, the development of new organic RTP materials with excellent processability, environmental friendliness, stability, and multiple functions is of significant importance to apply phosphorescent materials and protect the natural environment.

Herein, we chose natural cellulose as the raw material and introduced cyanomethylimidazolium cations (ImCN$^+$) and chloride anions (Cl$^-$) into the cellulose chain to obtain a cationic cellulose derivative: cellulose 1-cyanomethylimidazolium chloride (Cell-ImCNCl), which exhibited excellent RTP and processing properties (Fig. 1). The resultant Cell-ImCNCl can be easily processed into phosphorescent films, fibers, coatings, and patterns with antibacterial and water-resistant properties by using eco-friendly aqueous solution processing strategies.

## Results

**Synthesis and phosphorescence property.** Natural cellulose has many fascinating advantages, such as complete biodegradation, good biocompatibility, excellent mechanical properties, and sustainability. Moreover, numerous hydroxyl groups periodically arrange along the cellulose chain. The hydroxyl groups can form a strong hydrogen-bonding network, which effectively inhibits non-radiative transitions, and can be chemically modified with functional groups to promote the ISC and endow new functionality. The hydrogen-bonding interactions can restrict the movement of molecules, inhibit energy dissipation, reduce non-radiative transitions, and promote the return from triplet state T$_1$ to ground state S$_0$. The strategy of using the hydrogen-bonding interactions to achieve RTP has been widely used[51–54]. Clearly, cellulose is an ideal feedstock to prepare organic phosphorescent materials[55–60]. Therefore, we used cellulose as the input material to synthesize a series of cationic cellulose derivatives by a homogeneous modification process in the ionic liquid 1-allyl-3-methylimidazolium chloride (AmimCl) (Fig. 2a). In $^1$H-NMR spectra of the intermediate cellulose 2-chloropropionate (Cell-Cl) with different degrees of substitution (DS$_{Cl}$) (Supplementary Fig. 1a), the peak at 1.5–1.7 ppm is assigned to the methyl protons. In their FTIR spectra (Supplementary Fig. 1b), the carbonyl stretching vibration peak appears at 1741 cm$^{-1}$. Meanwhile, as DS$_{Cl}$ increases, the hydroxyl stretching vibration peak at 3450 cm$^{-1}$ shows a blue shift, and the intensity gradually weakens. When DS$_{Cl}$ reaches 3.0, the hydroxyl stretching vibration peak completely disappears. In the $^1$H-NMR spectra of the final product Cell-ImCNCl (Supplementary Fig. 2a), the new peaks at 9.7 ppm and 8.0 ppm are attributed to the protons of the imidazolium cation, and the new peak at 1.8 ppm is the methyl protons linked to the imidazolium cation. In the FTIR spectra (Supplementary Fig. 2b), the C≡N peak appears at 2065 cm$^{-1}$, and the characteristic peaks of the imidazolium cation appear at 1562 cm$^{-1}$ and 1174 cm$^{-1}$. These results evidently confirm that the cationic cellulose derivative Cell-ImCNCl was successfully prepared. By adjusting the synthesis conditions, it is easy to control the chemical structure of Cell-Cl and Cell-ImCNCl, such as DS$_{Cl}$ and DS of imidazolium cation (DS$_{CN}$). Because the ionic liquid AmimCl has the advanced characteristics of non-volatility, easy recyclability, good chemical inertness, and high polarity; the preparation process of phosphorescent Cell-ImCNCl is eco-friendly, large-scale, controllable, and highly efficient.

The introduction of cyanomethylimidazolium cations and chloride anions into cellulose promotes the ISC process, strengthens the hydrogen-bonding interactions and forms electrostatic attraction interactions, which effectively inhibit the

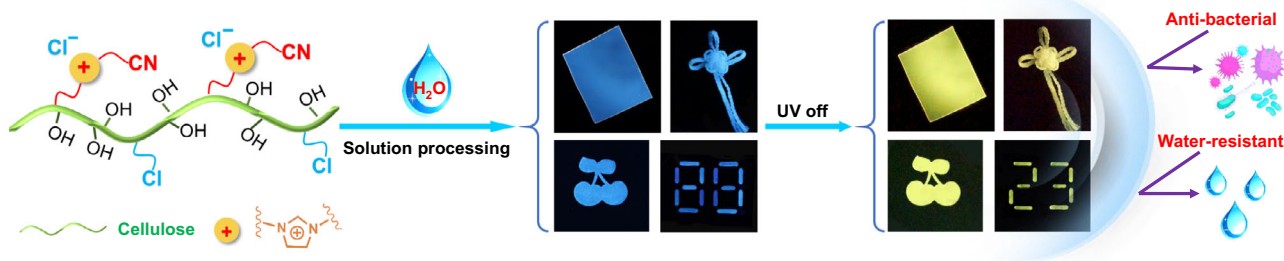

**Fig. 1 Phosphorescent Cell-ImCNCl.** Chemical structure, processability, and performance of the cationic cellulose derivative Cell-ImCNCl with RTP property.

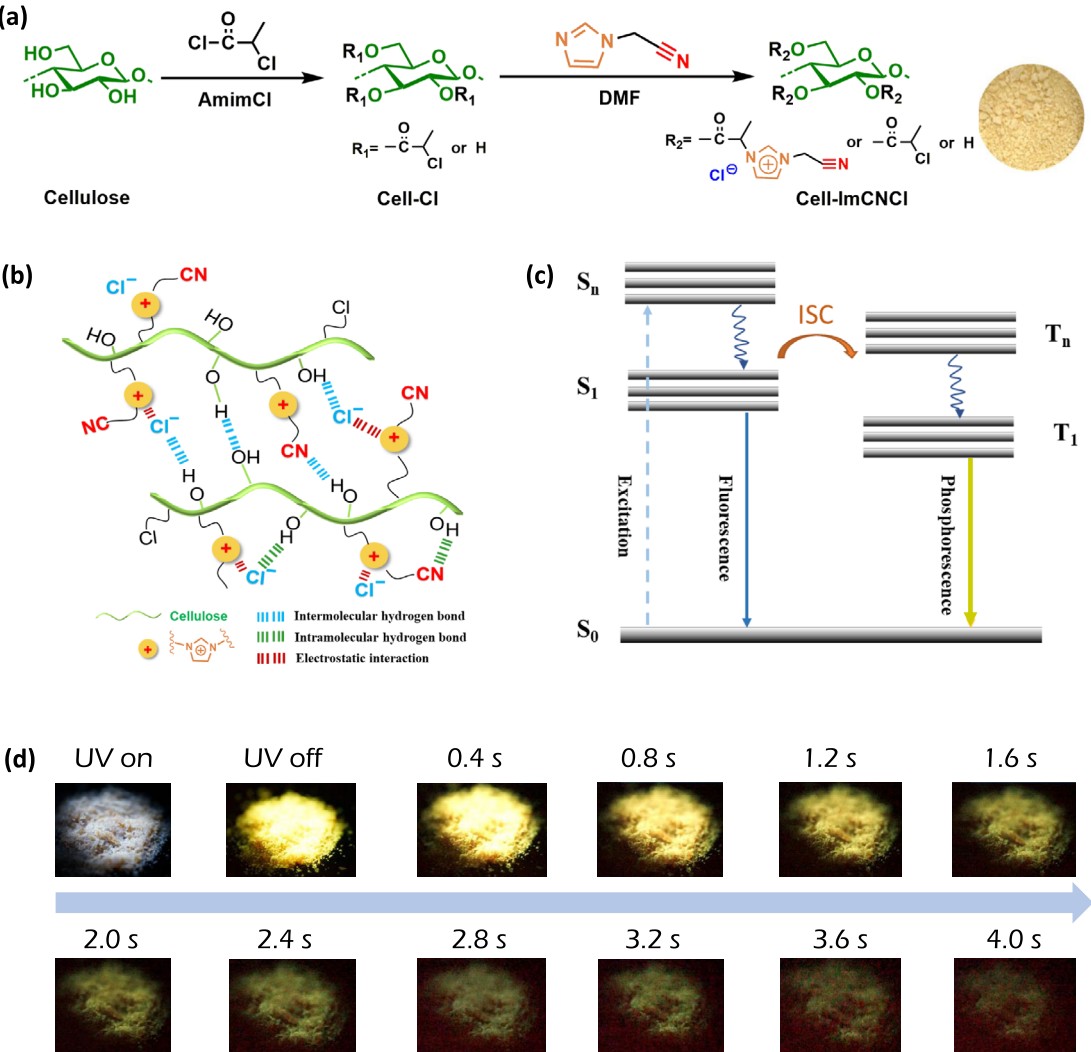

**Fig. 2 Synthesis, RTP mechanism, and performance of Cell-ImCNCl. a** Synthesis route and photograph of Cell-ImCNCl; **b** Schematic illustration of the RTP mechanism of Cell-ImCNCl; **c** Jablonski diagram energy level diagram of RTP materials; **d** Photographs of Cell-ImCNCl powder taken under 365 nm lamp and with the lamp off.

non-radiative transitions. Thus, amorphous Cell-ImCNCl shows excellent RTP performance (Fig. 2b, Supplementary Fig. 3). More specifically, in Cell-ImCNCl, the π-conjugated cyanomethylimidazolium cation acts as the luminophore. The cyano group and nitrogen on the imidazolium ring synergistically enhance the ISC process to promote the transition from excited state $S_1$ to triplet state $T_n$ (Fig. 2c, Supplementary Figs. 4 and 5). The strong hydrogen-bonding acceptors of the $Cl^-$ anion and the terminal cyano group on the imidazolium cation form strong hydrogen bond interactions with the strong hydrogen-bonding donors of the remaining hydroxyl groups on the cellulose chain. Meanwhile, the hydroxyl groups form hydrogen-bonding interactions with each other. The imidazolium cation and $Cl^-$ anion form electrostatic attraction interactions. These strong interactions limit the motions of Cell-ImCNCl to inhibit energy dissipation, stabilize the triplet state and promote the return from triplet state $T_1$ to ground state $S_0$, so phosphorescence emission is achieved at room temperature. After irradiation with a 365 nm ultraviolet lamp, the phosphorescence emission of Cell-ImCNCl continued for 4 s at room temperature, as observed by the naked eye (Fig. 2d).

**Phosphorescence mechanism**. To confirm the aforementioned phosphorescence emission mechanism, the 1-cyanomethylimidazolium chloride (CNMImCl) was synthesized and characterized firstly. The CNMImCl solid powder exhibits a similar phosphorescence emission to that of Cell-ImCNCl (Supplementary Fig. 4). Moreover, the CNMImCl aqueous solution (1 mg/mL) has a phosphorescence emission at 77–137 K, while gives a phosphorescence quenching at 197–273 K (Supplementary Fig. 5). These phenomena confirm that the imidazolium cation with the cyano group and nitrogen elements promotes the intersystem crossing. Further, we replaced 1-cyanomethylimidazole with 1-butylimidazole to prepare a new cationic cellulose derivative, cellulose 1-butylimidazolium chloride (Cell-BimCl) (Supplementary Fig. 6). Cell-BimCl has very poor phosphorescence performance, and its average phosphorescence lifetime is only 5 ms, which is much lower than the average phosphorescence lifetime of Cell-ImCNCl (158 ms). Compared with the cyanomethyl group, the butyl group in the butylimidazolium cation is a long alkyl segment with a lack of π-conjugated system, lone-pair electron and hydrogen bond forming ability, which weakens both ISC process and hydrogen-bonding interactions. Due to the lack of strong hydrogen-bonding interactions to fix the long

butyl segment, the triplet energy can be easily dissipated via the rotation and vibration of the butyl segment (Supplementary Fig. 7). Therefore, Cell-BimCl has a poor phosphorescence emitting performance. In addition, for 1-(cyanomethyl)imidazole, since there is no confinement effect originating from the polymer chains and strong interactions, the non-radiative transitions occur. As a result, 1-(cyanomethyl)imidazole has no phosphorescence (Supplementary Fig. 8). Besides, although cellulose has been reported to have a phosphorescence emission[55,58], cellulose has obviously weak phosphorescence performance in fact. The average phosphorescence lifetime is 5 ms and the photoluminescence quantum yield is 3.76% (Supplementary Fig. 9).

To further prove the above phosphorescence emission mechanism and to regulate the phosphorescence performance, we systematically studied the influence of the chemical structure of cationic cellulose derivatives on their RTP performance. As $DS_{CN}$ increases, the phosphorescence intensity, quantum yield, and average phosphorescence lifetime firstly increase and subsequently decrease (Fig. 3a–d). For Cell-ImCNCl with a total degree of substitution ($DS_t = DS_{Cl} + DS_{CN}$) of 1.24, when $DS_{CN}$ is 0.60, the phosphorescence intensity, quantum yield, and phosphorescence lifetime reach their maximum values. The maximum quantum yield and average phosphorescence lifetime are 11.81% and 158 ms, respectively. The increase in cyanomethylimidazolium cations promotes the ISC and strengthens the hydrogen-bonding interactions because the C≡N group is a strong hydrogen bond acceptor. The tightness degree between cellulose chains improves in Cell-ImCNCl. The non-radiative transitions are more strongly suppressed, so the phosphorescence intensity, quantum yield and phosphorescence lifetime are enhanced. However, when the content of cyanomethylimidazolium cations further increases, the electrostatic repulsion interactions between ions (cation–cation and anion–anion) dramatically increase. When $DS_{CN}$ exceeds the threshold, the electrostatic repulsion interactions exceed the sum of the hydrogen-bonding interactions and electrostatic attraction interactions because there are many ionic groups in the polymer chains. The stacking of polymer chains in Cell-ImCNCl becomes loose, and the non-radiative transitions increase, so the phosphorescence intensity, quantum yield, and phosphorescence lifetime decrease (Supplementary Fig. 10). Therefore, there is an optimal content of cyanomethylimidazolium cations to achieve the outstanding phosphorescence performance of Cell-ImCNCl.

As hydrogen-bonding donors, the hydroxyl groups in Cell-ImCNCl play an essential role in the phosphorescence emission. Combine the discussion in the previous section, we kept $DS_{CN}$ with ~0.60 and changed $DS_t$ to regulate the content of hydroxyl groups in Cell-ImCNCl. When $DS_t$ increases, which means a decrease in the content of hydroxyl groups, the phosphorescence intensity, quantum yield and phosphorescence lifetime firstly increase and subsequently decrease (Fig. 3e-h). For example, when $DS_{CN}$ is ~0.60, Cell-ImCNCl with a $DS_t$ of 1.24 has the highest phosphorescence performance in our current work. $DS_{CN}$ remains unchanged, and $DS_t$ increases, so $DS_{Cl}$ increases in Cell-ImCNCl. The increase in the strong hydrogen bond acceptor chlorine can enhance the hydrogen-bonding interactions in Cell-ImCNCl to inhibit the non-radiative transitions more effectively, so the phosphorescence performance increases. However, when $DS_t$ increases, the hydroxyl groups acting as the hydrogen bond donors decrease. Therefore, when $DS_t$ exceeds the threshold, the hydrogen-bonding interactions in Cell-ImCNCl will weaken and the non-radiative transitions will be enhanced. The phosphorescence intensity, quantum yield, and phosphorescence lifetime will decrease. Thus, the regulation of the hydroxyl group content in Cell-ImCNCl is of significant importance to adjust the phosphorescence performance.

The Cl⁻ anion in Cell-ImCNCl is also a strong hydrogen-bonding acceptor, and can form strong hydrogen-bonding interactions with hydroxyl groups. When the Cl⁻ anion is replaced with other anions, the hydrogen-bonding interactions in the system will change. For example, the cellulose 1-cyanomethylimidazolium fluoride (Cell-ImCNF) with negligible heavy atom effect and strong hydrogen-bonding basicity of fluoride ion exhibits a similar phosphorescence quantum yield and phosphorescence lifetime to those of Cell-ImCNCl (Supplementary Fig. 11), while the cellulose 1-cyanomethylimidazolium bromide (Cell-ImCNBr) (Supplementary Fig. 12) with strong heavy atom effect and weak hydrogen-bonding basicity of bromide ion gives a significantly decreased phosphorescence intensity, quantum yield, and phosphorescence lifetime (Fig. 3j-l). After an anion exchange process, we replaced the Cl⁻ anion with anions with weaker hydrogen bond acceptor abilities, such as $BF_4^-$, $PF_6^-$, $Tf_2N^-$, $C(CN)_3^-$, $N(CN)_2^-$, and SCN⁻ (Fig. 3i, Supplementary Fig. 13). As a result, the hydrogen-bonding interactions in the resultant amorphous Cell-ImCNX weaken (Supplementary Fig. 14), and they have reduced phosphorescence performance compared with Cell-ImCNCl (Fig. 3j-l). These results indicate that the heavy atom effect of the anions has a negligible effect on the phosphorescence properties of cationic cellulose derivatives. In contrast, the strong hydrogen-bonding capability of the anions is a determining factor for the phosphorescence emission, because the strong hydrogen-bonding interactions inhibit the non-radiative transitions.

The above results demonstrate the phosphorescence emission mechanism. The imidazolium cations with the cyano group and nitrogen element in Cell-ImCNCl promote the ISC process. The cyano-containing cations, chloride anions, and hydroxyl groups interact with one another via hydrogen-bonding interactions and electrostatic attraction interactions. The non-radiative transitions are effectively inhibited, so phosphorescence emission is achieved at room temperature. Adjusting the chemical structure of the cationic cellulose derivatives changes the hydrogen-bonding interactions and electrostatic interactions, which causes a change in phosphorescence performance. Cell-ImCNCl with a $DS_t$ of 1.24 and a $DS_{CN}$ of 0.60 exhibits the optimal phosphorescence performance.

**Phosphorescence materials**. Phosphorescent Cell-ImCNCl has excellent water solubility because there are numerous ionic groups along the polymer chain. By using water as the solvent and employing various solution processing methods, we can process Cell-ImCNCl into different material forms, including phosphorescent films, fibers, coatings, patterns, etc. (Fig. 4). For example, by using a facile doctor blade coating method, a flexible phosphorescent film was obtained (Fig. 4a, b). By employing a facile dip coating method, phosphorescent PVA fibers and phosphorescent cellulose films were obtained (Fig. 4a, Supplementary Fig. 15). The phosphorescent PVA fibers could be woven into phosphorescent "Chinese knots" (Fig. 4c). Moreover, via the same method, a large roll of phosphorescent cellulose fibers, large-area phosphorescent cellulose films, and numerous phosphorescent microspheres have been obtained easily (Supplementary Figs. 16 and 17). Via screen printing, inkjet printing, and mask casting methods, phosphorescent patterns are formed on various substrates, including paper, ceramics, glass, plastic, stainless steel, and aluminum foil (Fig. 4a, d–i, Supplementary Fig. 15c). Such excellent processability and formability make phosphorescent Cell-ImCNCl applicable in many fields, such as complex anticounterfeiting, information encryption and storage, smart labels, packaging, special fibers, and detection sensors. In particular, because the solvent is only water, the entire processing process is beneficial to environmental protection.

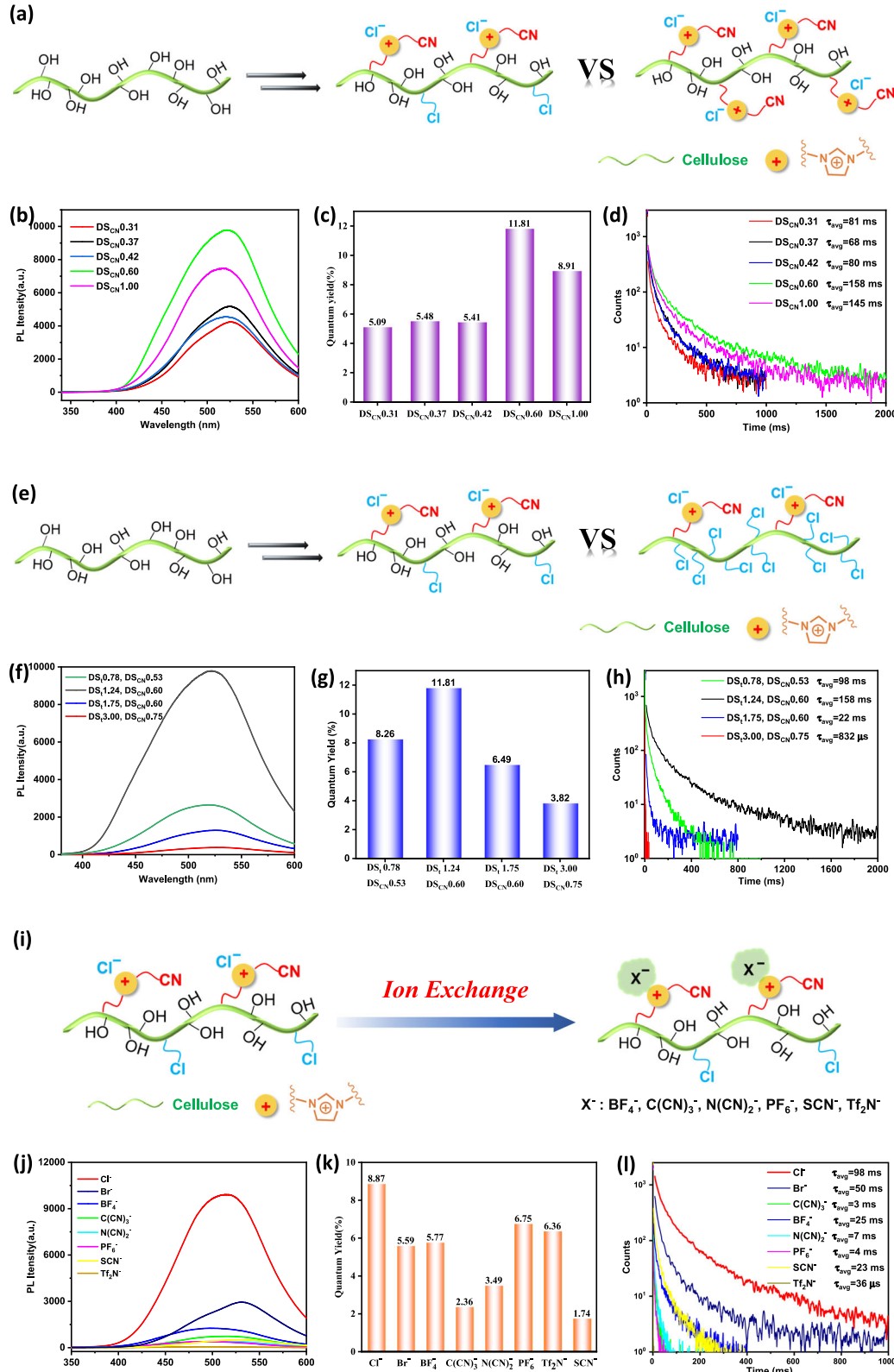

**Fig. 3 Influence of the chemical structure of Cell-ImCNCl on its RTP performance. a** Schematic diagram of Cell-ImCNCl with different DS_CN; **b** RTP spectra of Cell-ImCNCl with different DS_CN (Ex = 320 nm); **c** Photoluminescence quantum yield of Cell-ImCNCl with different DS_CN (Ex = 320 nm); **d** RTP lifetime spectra of Cell-ImCNCl with different DS_CN; **e** Schematic diagram of Cell-ImCNCl with different DS_t; **f** RTP spectra of Cell-ImCNCl with different DS_t (Ex = 320 nm); **g** Photoluminescence quantum yield of Cell-ImCNCl with different DS_t (Ex = 320 nm); **h** RTP lifetime spectra of Cell-ImCNCl with different DS_t; **i** Schematic diagram of Cell-ImCNX with different anions; **j** RTP spectra of Cell-ImCNX (Ex = 320 nm); **k** Photoluminescence quantum yield of Cell-ImCNX with different anions; **l** RTP lifetime spectra of Cell-ImCNX with different anions.

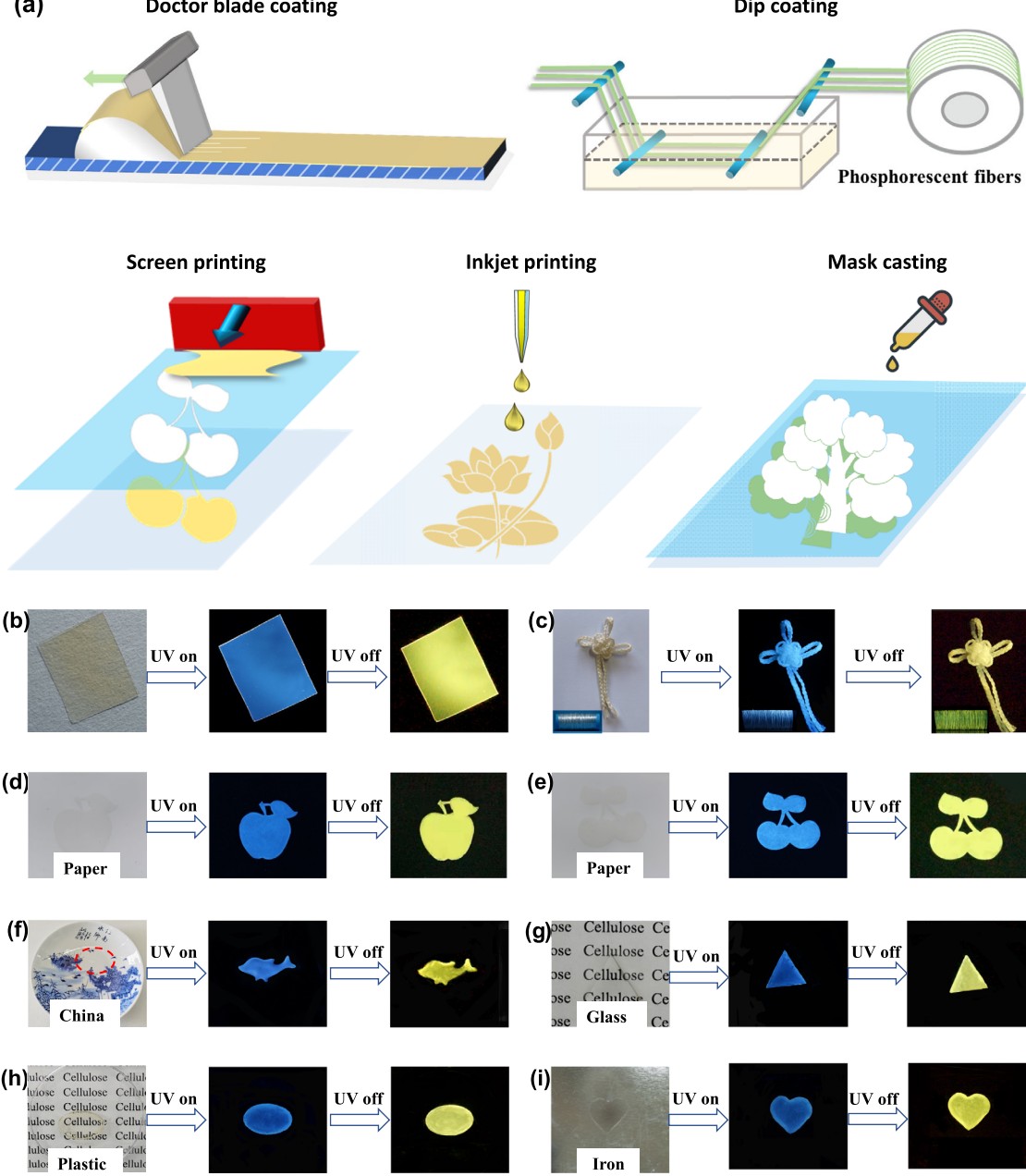

**Fig. 4 Processability and formability of Cell-ImCNCl. a** Processing methods of Cell-ImCNCl; **b** Phosphorescent Cell-ImCNCl film made by the doctor blade coating method; **c** Phosphorescent "Chinese knot" from PVA fibers produced by the dip coating method; **d–i** Phosphorescent patterns on various substrates. The ultraviolet light used is 365 nm.

**Water-resistant phosphorescence patterns**. Although it is obviously environmentally friendly to use water as the solvent for processing, the stability of the obtained materials must be considered in practical applications. Taking advantage of the remaining hydroxyl groups on the cellulose chain, we used a small amount of glutaraldehyde as the crosslinking agent to construct a chemical crosslinking during the processing process. Due to the formation of physical and chemical double crosslinking, the resultant phosphorescent materials were water-resistant (Fig. 5a). With this strategy, the Cell-ImCNCl/glutaraldehyde aqueous solution can be used to obtain phosphorescent and water-resistant patterns and coatings via a spray coating technique. For example, we made a "plane" pattern on a glass substrate. The "plane" pattern showed excellent phosphorescence and water resistance (Fig. 5b, c). After being immersed into water for 24 h, the "plane" pattern had a negligible change. In addition, the antibacterial ring is formed around the phosphorescent pattern, and the obtained phosphorescent pattern had a killing effect on both Gram-negative bacteria (Escherichia coli) and Gram-positive bacteria (Staphylococcus aureus) with restricted growth (Fig. 5d). The outstanding antibacterial property probably originates from a high ion concentration environment around the phosphorescent pattern because the dual crosslinking structure limits the movement of polymer chains and ions. The cationic groups interact with the bacteria with negatively charged surface to disturb the bacterial cell membrane through terminal alkane chains and ultimately lead to the death of bacteria[61–66].

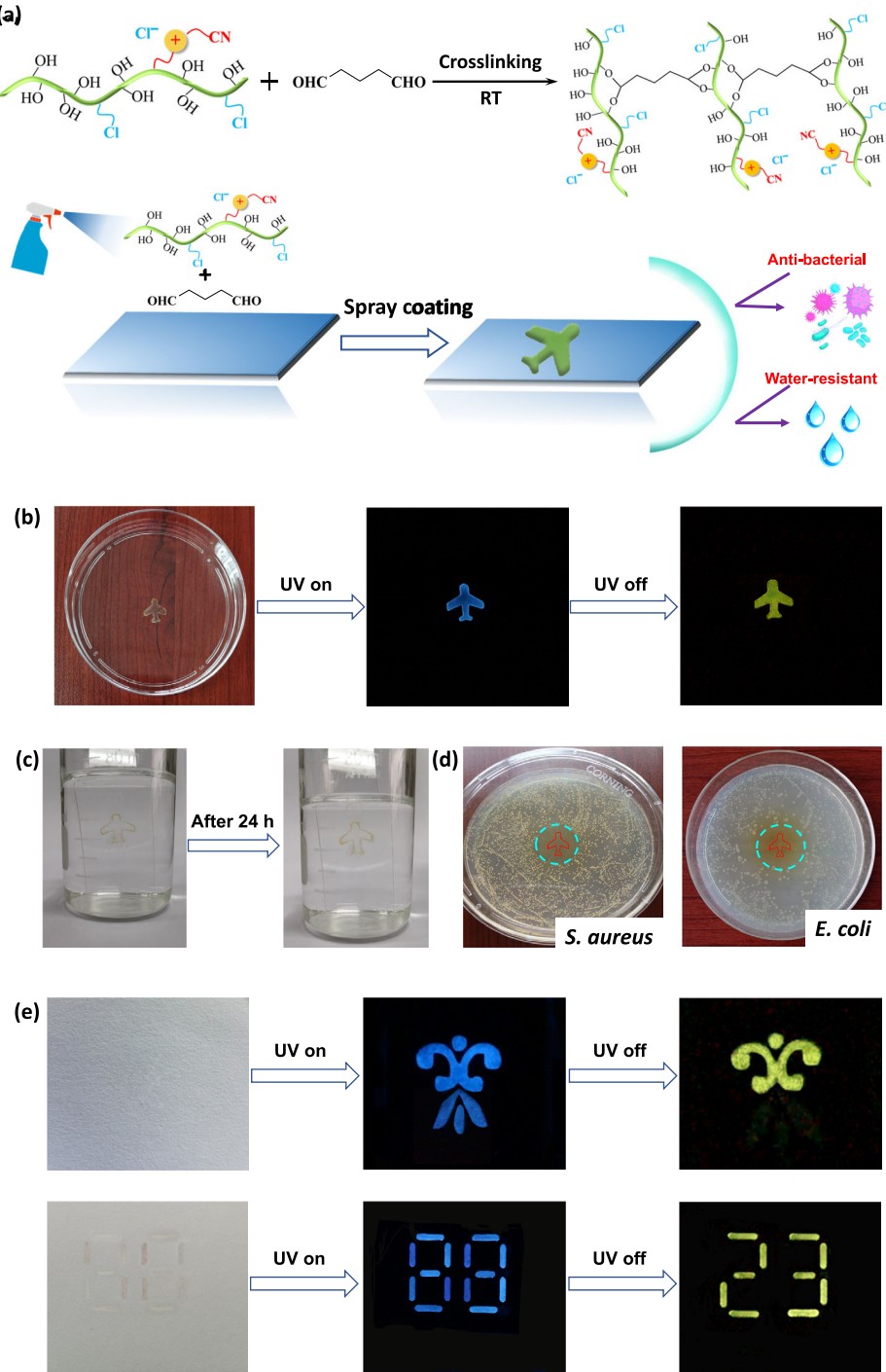

**Fig. 5 Water-resistant phosphorescent materials based on Cell-ImCNCl. a** Schematic diagram of the preparation process of water-resistant phosphorescent materials; **b** Optical images of the water-resistant phosphorescent pattern; **c** Water-resistant test of the phosphorescent patterns; **d** Antibacterial properties of water-resistant phosphorescent patterns by the inhibition ring experiment (*Staphylococcus aureus* (*S. auresus*) and *Escherichia coli* (*E. coli*)); **e** Anticounterfeiting patterns made by phosphorescent Cell-ImCNCl and fluorescent Cell-BimCl. The ultraviolet light used is 365 nm.

In brief, Cell-ImCNCl, which has excellent phosphorescence properties, water solubility, and processability, can be easily processed into phosphorescent, water-resistant and antibacterial patterns, coatings and bulk materials via a facile, and eco-friendly process. Considering the excellent biodegradability and sustainability of cellulosic materials, the resulting cellulose-based RTP materials are environmentally friendly. Thus, they are expected to be applicable in many fields, such as complex anticounterfeiting, information security and encryption, disposable smart labels,

food and drug storage, and monitoring. For example, we used phosphorescent Cell-ImCNCl and fluorescent Cell-BimCl, which have similar structures and good compatibility, to print the patterns for anticounterfeiting and information encryption (Fig. 5e). The pattern gives an image under ultraviolet light, and it becomes another image after the ultraviolet light is turned off. As shown in Fig. 5e, the number is "88" under ultraviolet light. When the ultraviolet light is turned off, the number becomes "23".

## Discussion

Using the strong hydrogen-bonding capability and facile chemical modification characteristics of natural cellulose due to abundant hydroxyl groups, we introduced cyanomethylimidazolium cations ($ImCN^+$) and chloride anions ($Cl^-$) into the cellulose chain to obtain the cationic cellulose derivative Cell-ImCNCl with excellent processing and RTP properties. The degrees of substitution, anions, and cations significantly affected RTP performance. The imidazolium cations with the cyano group and nitrogen element promoted ISC. Moreover, the cyano-containing cations, chloride anions, and remaining hydroxyl groups formed strong hydrogen-bonding interactions and electrostatic attraction interactions, which effectively suppressed the non-radiative transitions. Adjusting the chemical structure of Cell-ImCNCl could control the phosphorescence lifetime and photoluminescence quantum yield, which were up to 158 ms and 11.81%, respectively. More fascinatingly, phosphorescent Cell-ImCNCl was easily and quickly dissolved in water, so it is very convenient to fabricate phosphorescent films, fibers, coatings, and patterns with Cell-ImCNCl as the raw material and by employing various solution processing methods. A small amount of glutaraldehyde was added to the Cell-ImCNCl solution to form a stable double crosslinking structure. As a result, the resulting phosphorescent patterns exhibited excellent antibacterial properties and water resistance. In summary, we have developed a simple and practical method to prepare organic RTP materials, which are easy-to-process, antibacterial, water-resistant, and environmentally friendly. The multi-functional Cell-ImCNCl is expected to be used in advanced anticounterfeiting, information security and encryption, disposable smart labels, and the preservation and monitoring of food and medicine.

## Methods

**Materials**. Cellulose (microcrystalline cellulose, PH-101) with an average degree of polymerization (DP) of 220 was purchased from Beijing Fengli Jingqiu Commerce and Trade Company (China). It was dried in vacuum at 80 °C for 24 h. 2-Chloropropionyl chloride, 2-bromopropionyl chloride, (1-cyanomethyl) imidazole, 1-butylimidazole, 1-methylimidazole, allyl chloride, lithium tetrafluoroborate ($LiBF_4$), sodium hexafluorophosphate ($NaPF_6$), lithium tetrafluoroborate ($LiTf_2N$), sodium tricyanomethanide ($NaC(CN)_3$), sodium dicyanamide ($Na(CN)_2$), sodium thiocyanate (NaSCN), and glutaraldehyde were purchased from Innochem and J&K Scientific. Ultradry N,N'-dimethylformamide (DMF) with a moisture content below 30 ppm was obtained from Innochem. The above chemicals were analytical grade, and were used without further purification. Chromatography N,N'-dimethylformamide (DMF), acetone, and ethanol were received from Tianjin Concord Technology Co., Ltd.

**Synthesis of AmimCl**. 1-Methylimidazole (400 mL) and allyl chloride (700 mL) at a molar ratio 1:1.1 were added to a round-bottomed flask fitted with a reflux condenser for 12 h at 50 °C with stirring. The unreacted chemical reagents and other impurities, such as water, were removed by vacuum distillation. The obtained AmimCl is slightly yellow. The purity of AmimCl is 99.7%, which is determined by a high-performance liquid chromatography method with water as mobile phase. The water content in AmimCl determined by Karl Fischer method was less than 0.3 wt%. $^1$H-NMR (400 MHz, DMSO-$d_6$): δ 9.10 (s, 1H), 7.72 (s, 1H), 7.70 (s, 1H), 6.04 (m, 1H), 5.39 (m, 2H), 4.46 (d, 2H), 3.89 (s, 3H); IR (Nujol): 3051 cm$^{-1}$ (=C–H), 1646 cm$^{-1}$ (C=C), 1571 cm$^{-1}$ (C=N), 1425 cm$^{-1}$ (=CH$_2$), 1171 cm$^{-1}$ (C–N).

**Synthesis of cellulose 2-chloropropionate (Cell-Cl)**. Three grams (18.52 mmol) of cellulose was completely dissolved in 114 g of the AmimCl at 80 °C. Subsequently, 10 mL of ultradry DMF was added to the cellulose/AmimCl solution. Then, 2-chloropropionyl chloride (2.35–11.76 g, 18.52–92.59 mmol) was added to the cellulose/AmimCl/DMF solution under ice bath conditions. After 2 min of stirring, the reaction system was transferred into an oil bath at 40 °C for 0.5–2 h. The product was precipitated in a mixed solvent of ethanol/water (v/v = 1/1) and collected by filtration. After being washed three times with ethanol, the product was redissolved in dimethyl sulfoxide (DMSO), precipitated again in ethanol/water (v/v = 1/1) and washed three more times. Finally, the product was filtered and dried in vacuum at 60 °C for 24 h before characterization. $^1$H-NMR (400 MHz, DMSO-$d_6$): δ 2.80–5.50 (m, 8H), 1.60 (s, 3H); IR (Nujol): 3452 cm$^{-1}$ (O–H), 1739 cm$^{-1}$ (C=O).

**Synthesis of cellulose 1-cyanomethylimidazolium chloride (Cell-ImCNCl)**. The intermediate Cell-Cl (DS$_{Cl}$ = 1.24, 0.5 g, 2.2 mmol) was dissolved in DMF. Then, 1-cyanomethylimidazole (0.6–2.6 g, 5.4–24.2 mmol) was added to the Cell-Cl/DMF solution at 80 °C for 24–48 h. The product was precipitated in acetone, and collected by filtration. The crude product was washed with acetone three times. Finally, the product was dried in vacuum at 60 °C for 24 h before characterization. $^1$H-NMR (400 MHz, DMSO-$d_6$): δ 9.47 (s, 1H), 7.98 (s, 2H), 6.00–2.98 (m, 10H), 1.81 (s, 3H), 1.60 (s, 3H); IR (Nujol): 3417 cm$^{-1}$ (O–H), 2075 cm$^{-1}$ (C≡N), 1751 cm$^{-1}$ (C=O), 1568 cm$^{-1}$ (C=N), 1176 cm$^{-1}$ (C–N).

**Synthesis of Cell-ImCNX**. Two hundred milligrams (0.59 mmol) of Cell-ImCNCl (DS$_t$ = 1.30, DS$_{CN}$ = 0.53) was dissolved in 5 mL of ultrapure water. Then, NaPF$_6$ (526 mg), LiTf$_2$N (896 mg), NaC(CN)$_3$ (353 mg), Na(CN)$_2$ (280 mg), NaSCN (253 mg), or LiBF$_4$ (293 mg) was added to the Cell-ImCNCl aqueous solution which was stirred for 30 min. The solution was centrifuged, and the solid product was washed with ultrapure water three times. Finally, the product was dried in vacuum at 60 °C for 48 h before characterization. $^1$H-NMR (400 MHz, DMSO-$d_6$): δ 9.47 (s, 1H), 7.98 (s, 2H), 6.00–2.98 (m, 10H), 1.81 (s, 3H), 1.60 (s, 3H); IR (Nujol): 3417 cm$^{-1}$ (O–H), 2075 cm$^{-1}$ (C≡N), 1751 cm$^{-1}$ (C=O), 1568 cm$^{-1}$ (C=N), 1176 cm$^{-1}$ (C–N).

**Synthesis of Cell-BimCl**. The intermediate product Cell-Cl (DS$_{Cl}$ = 1.24, 700 mg, 2.6 mmol) was dissolved in 15 mL of DMF. Then, 1-butylimidazole (613 mg, 4.9 mmol) was added to the Cell-Cl/DMF solution. The reaction was conducted at 80 °C for 24 h. The product was precipitated in acetone, and collected by filtration. The crude product was washed with acetone three times. The product was dried in vacuum at 60 °C for 24 h before characterization. $^1$H-NMR (400 MHz, DMSO-$d_6$): δ 9.40 (s, 1H), 7.84 (s, 2H), 5.90–2.75 (m, 10H), 1.79 (s, 3H), 1.61 (s, 3H), 1.26 (s, 2H), 0.89 (s, 3H); IR (Nujol): 3338 cm$^{-1}$ (O–H), 2973 cm$^{-1}$ (C–H), 1745 cm$^{-1}$ (C=O), 1560 cm$^{-1}$ (C=N), 1171 cm$^{-1}$ (C–N).

**Synthesis of cellulose 2-bromopropionate (Cell-Br)**. Three grams (18.52 mmol) of cellulose was completely dissolved in 114 g of the ionic liquid AmimCl at 80 °C. Then, 2-bromopropionyl chloride (16.32 g, 7.6 mmol) was added to the cellulose/AmimCl solution under ice bath conditions. After 2 min of stirring, the reaction system was transferred into an oil bath at 40 °C for 18 min. The product was precipitated in a mixed solvent of ethanol/water (v/v = 1/1) and collected by filtration. After being washed three times with ethanol, the product was redissolved in dimethyl sulfoxide (DMSO), precipitated again in ethanol/water (v/v = 1/1) and washed three times. Finally, the product was filtered and dried in vacuum at 60 °C for 24 h before characterization. $^1$H-NMR (400 MHz, DMSO-$d_6$): δ 2.85–5.30 (m, 7H), 1.61 (s, 3H); IR (Nujol): 3494 cm$^{-1}$ (O–H), 1747 cm$^{-1}$ (C=O).

**Synthesis of cellulose 1-cyanomethylimidazolium bromide (Cell-ImCNBr)**. The intermediate product Cell-Br (DS$_{Br}$ = 1.30, 545 mg, 1.9 mmol) was dissolved in 15 mL of DMF. Then, 1-cyanomethylimidazole (830 mg, 7.6 mmol) was added to the Cell-Br/DMF solution. The reaction was conducted at 80 °C for 24 h. The product was precipitated in acetone and collected by filtration. The crude product was washed with acetone three times. Finally, the product was dried in vacuum at 60 °C for 24 h before characterization. $^1$H-NMR (400 MHz, DMSO-$d_6$): δ 9.49 (s, 1H), 7.98 (s, 2H), 6.00–2.75 (m, 10H), 1.81 (s, 3H), 1.61 (s, 3H); IR (Nujol): 3438 cm$^{-1}$ (O–H), 2075 cm$^{-1}$ (C≡N), 1751 cm$^{-1}$ (C=O), 1562 cm$^{-1}$ (C=N), 1174 cm$^{-1}$ (C–N).

**Preparation of water-resistant phosphorescent pattern**. First, 200 μL of 10% glutaraldehyde aqueous solution was added to 1 mL of Cell-ImCNCl aqueous solution (100 mg/mL). Then, the solution was sprayed on a glass plate covered by a "Plane" mold. After drying, the water-resistant phosphorescence "Plane" pattern was obtained.

**Measurements**. $^1$H-NMR spectra were acquired on a Bruker AV-400 NMR spectrometer with 16 scans at room temperature in DMSO-$d_6$. A 20 μL aliquot of CF$_3$COOH-$d_1$ was added to shift the signals of the free hydrogens downfield. The degree of substitution (DS) of cellulose esters was calculated directly from $^1$H-NMR by the following equation.

$$DS = \frac{7I_{Ester}}{nI_{AGU}}$$

where, $I_{Ester}$ is peak integral of ester groups; $I_{AGU}$ is peak integral of protons of anhydroglucose unit; $n$, the number of protons on the ester groups.

The FTIR spectra of solid samples (32 scans at 0.5 cm$^{-1}$ digital resolution) were recorded with a Nicolet 6700 FTIR spectrometer (Thermo Fisher, USA) from 650 to 4000 cm$^{-1}$ at a resolution of 4 cm$^{-1}$. The KBr disc technique was employed, and the mass fraction of the tested sample was about 2 wt%. X-ray photoelectron spectroscopy (XPS) was performed on a Thermo Scientific ESCALab 250Xi spectrometer using 200 W monochromatic Al Kα radiation. The 500 μm X-ray spot was used for XPS analysis.

Fluorescence spectra and phosphorescence spectra were recorded with a Hitachi F-7000 fluorescence spectrophotometer. The photoluminescence quantum yield and phosphorescence lifetime were measured on an Edinburgh FLS980 steady-state transient fluorescence spectrometer with an integrating sphere and a microsecond flash lamp. The phosphorescence lifetime was measured by multi-channel single photon technology. Fluorescence and phosphorescence images were captured with a digital camera (SONY α7, Japan). The X-ray diffractograms (XRD) were recorded in reflection mode in the angular range of 5–50° (2θ) with a scanning speed of 5°/min by using an X-ray diffractometer (D/MAX-2500, Rigaku Denki, Japan).

**Antibacterial performance test**. The agar medium (5 g of peptone, 3 g of beef extract, 5 g of NaCl, 15 g of agar, pH = 7.0) was sterilized under a pressure of 15 lbs for 30 min and transferred to a sterile petri dish for solidification. Then, 0.1 mL Staphylococcus aureus or Escherichia coli suspension (106–107 cfu/mL) was evenly spread on an agar plate and naturally dried. All operations were performed under aseptic conditions. The sample was placed on the agar plate and incubated upside down in a 37 °C culture room for 20 h. Finally, the inhibition ring was observed.

**Reporting summary**. Further information on research design is available in the Nature Research Reporting Summary linked to this article.

## Data availability

All relevant data are included in this article and its Supplementary Information files.

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

## Acknowledgements

This work was supported by the National Natural Science Foundation of China (No. 51773210) (J.Z.) and Youth Innovation Promotion Association CAS (No. 2018040) (J.M.Z.).

## Author contributions

X.Z., J.M.Z., and J.Z. conceived the idea. X.Z. performed the experiments. Y.H.C., J.X.Y., and C.C.Y. offered help to X.Z. for the experiments. X.Z., J.M.Z., and J.Z. discussed the results and wrote the manuscript.

## Competing interests

The authors declare no competing interests.
