## [Peer Review File · Nature Communications]

REVIEWER COMMENTS

Reviewer #1 (Remarks to the Author):

In this manuscript entitled “Ultralong Phosphorescence Cellulose with Excellent Anti-bacterial, Water-resistant and Ease-to-process Performance”, Zhang et al. presented a phosphorescent cationized cellulose derivative with properties that make it easy to process by simply introducing ionic structures into cellulose chains. The resultant cellulose based RTP material can be easily processed into phosphorescent films, fibers, coatings and patterns by using environmentally friendly aqueous solution processing strategies. The study is an interesting research but lacks details research in term of mechanism understanding or real applications. Some comments are listed below for your information

1. For your system, cellulose itself needs to be used as a reference, those data have not been presented
2. For your system, what are the crystallinity changes when various of ionic structures introduced into the cellulose?
3. For your system, do you know the purity of the raw materials, does it affect the RTP?
4. Try to cite some references on hydrogen bonds effects on RTP
5. The propose of using glutaraldehyde in the system as a crosslinking agent need to be further clarified, did you check the crystallinity, FTIR and others, also how about the purity of the system?
6. Antibacterial properties and the explanation is too brief, more evidence and reference needed to support your claim

The authors need to choose focusing on mechanism study or application study, in either way, the authors need to put strong evidence to support your claim.

Reviewer #2 (Remarks to the Author):

The manuscript is well written and describes very thoroughly the synthesis processes and the applications of an ultralong phosphorescence cellulose. The manuscript includes very interesting properties of the synthesized material, which corroborates the versatility of the applications at which the material can be used. However, even though the synthesis of the material is somewhat innovative, the new synthesis and the long phosphorescence exhibited by the cellulose based materials does not meet the novelty required for a publication in a journal such as Nature Communications. There are other journals specialized in materials where this manuscript would be a better suit. Just as examples, there are several research groups that have published similar materials in recent years, some of which have been cited by the authors in references 51-53. Therefore, my recommendation is to seek a more specialized journal to publish the work due to an insufficient novelty of the work to be published in Nature Communications.

Reviewer #3 (Remarks to the Author):

In this work, the authors reported cellulose-based easy-to-process RTP materials, which can be applied as antibacterial, water-resistant, and eco-friendly phosphorescent films, fibers, coatings, and patterns. The multiple hydrogen-bonding interactions and electrostatic attraction interactions of the RTP cellulose have efficiently suppressed the non-radiative transitions to achieve ultralong phosphorescence emission. Although this work has some merits, the novelty is rather limited and the data in the present are not convincing. Below are my concerns and suggestions.

Major concerns:

1. To achieve phosphorescence emission, the authors chose natural cellulose as the raw material and introduced cyanomethylimidazolium cations (ImCN⁺) and chloride anions (Cl⁻) into the cellulose chain to obtain the RTP materials. However, cellulose for sustainable phosphorescent materials has already been reported (Carbon 171 (2021) 946e952; ACS Nano 2020, 14, 11130–11139). Therefore, the novelty of this work is limited.
2. In page 7, the author increased degree of substitution of imidazolium cation (DSCN). It causes efficiency and lifetime of phosphorescence enhanced first and then decreased. However, when the DSCN is 0.37, the corresponding efficiency and lifetime contradict the laws proposed by author. Why could the unique phenomenon be observed in 0.37?
3. In page 8. The author kept DSCN and changed DSt, but the experiment results of Fig. 3h lacks of regularity. Therefore, the author proposed the most appropriate value of 1.24 for DSt is probably not accurate.
4. The authors made a "plane" with Cell-ImCNCl/glutaraldehyde on a glass substrate, after being immersed into water for 24 hours; the "plane" pattern had a negligible change. The authors demonstrated that the Cell-ImCNCl has high water-resistant property. However, whether the material changes after immersed into water requires more microscopic observation such as observation by microscope or scanning electron microscope. Besides, more rigorous testing methods such as the water contact angle of Cell-ImCNCl also should be carried out to prove water-resistant.
5. The authors detected the antibacterial properties of phosphorescent pattern by LB agar colony in Figure 5d and elucidated that the excellent antibacterial ability probably originated from a high ion concentration environment around the phosphorescent pattern. While, the phosphorescent pattern is composed of Cell-ImCNCl/glutaraldehyde aqueous solution, it is well known that glutaraldehyde is

commonly used for biological fixation and is highly lethal to bacteria. Therefore, the authors should rule out the effect of glutaraldehyde on antibacterial activity.

6. The authors clarify that the obtained organic RTP material has much excellent properties such as easy-to-process, antibacterial, water-resistant, and environmentally-friendly. What role do these features play in the specific application, or how do they relate to each other? It should be further elaborated.

Other comments:

1. The abbreviation of a noun should be given its full name when it first appears. Such as, Page 12, lines 266 and 267, *Escherichia coli* (*E. coli*) and *Staphylococcus aureus* (*S. aureus*). The name of bacteria should be in italics.

2. The description of the figure annotations should be more detailed and easier for readers to understand. For example, Fig. 5d, the antibacterial assay method should be included.

Answers to Comments by Reviewers

Answers to Comments by Reviewer # 1

- (1) **Reviewer #1 wrote:** *In this manuscript entitled “Ultralong Phosphorescence Cellulose with Excellent Anti-bacterial, Water-resistant and Ease-to-process Performance”, Zhang et al. presented a phosphorescent cationized cellulose derivative with properties that make it easy to process by simply introducing ionic structures into cellulose chains. The resultant cellulose based RTP material can be easily processed into phosphorescent films, fibers, coatings and patterns by using environmentally friendly aqueous solution processing strategies. The study is **interesting research** but lacks details research in term of mechanism understanding or real applications.*

Answer: Thanks for your nice comments. We have added the experiments and discussion to reveal the phosphorescence mechanism and further demonstrate the real applications in the revised manuscript as suggestion.

In order to understand the mechanism clearly, the 1-cyanomethylimidazolium chloride (CNMImCl) was synthesized and characterized firstly. The CNMImCl solid powder exhibits a similar phosphorescence emission to that of Cell-ImCNCl (Fig. R1). Moreover, the CNMImCl aqueous solution (1 mg/mL) has a phosphorescence emission at 77-137 K, while gives a phosphorescence quenching at 197-273 K (Fig. R2). These phenomena confirm that the imidazolium cation with the cyano group and nitrogen element promotes the intersystem crossing (ISC). Further, we replaced 1-cyanomethylimidazole with 1-butyylimidazole to prepare a new cationic cellulose derivative, cellulose 1-butyylimidazolium chloride (Cell-BimCl) (Fig. S6). Cell-BimCl has very poor phosphorescence performance, and its average phosphorescence lifetime is only 5 ms, which is much lower than the average phosphorescence lifetime of Cell-ImCNCl (158 ms). Compared with the cyanomethyl group, the butyl group in the butylimidazolium cation is a long alkyl segment with a lack of π -conjugated system, lone-pair electron and hydrogen bond forming ability, which weakens both ISC process and hydrogen bonding interactions. Due to the lack of strong hydrogen bonding interactions to fix the long butyl segment, the triplet energy can be easily dissipated via the rotation and vibration of the butyl segment (Fig. S7). Therefore, Cell-BimCl has a poor phosphorescence emitting performance. In addition, for

1-(cyanomethyl)imidazole, since there is no confinement effect originating from the polymer chains and strong interactions, the non-radiative transitions occur. Therefore, 1-(cyanomethyl)imidazole has no phosphorescence (Fig. S8).

Furthermore, the cellulose 1-cyanomethylimidazolium fluoride (Cell-ImCNF) with negligible heavy atom effect and strong hydrogen-bonding basicity of fluoride ion exhibits a similar phosphorescence quantum yield and phosphorescence lifetime to Cell-ImCNCl (Fig. R3), while the cellulose 1-cyanomethylimidazolium bromide (Cell-ImCNBr) with strong heavy atom effect and weak hydrogen-bonding basicity of bromide ion gives a significantly decreased phosphorescence quantum yield and lifetime (Fig. 3k and 3l). If the Cl⁻ anion was replaced with the anions with weaker hydrogen bond acceptor abilities, such as BF₄⁻, PF₆⁻, Tf₂N⁻, C(CN)₃⁻, N(CN)₂⁻, SCN⁻ (Fig. 3i). As a result, the resultant new cationic cellulose derivatives Cell-ImCNX have reduced phosphorescence performance compared with Cell-ImCNCl (Fig. 3j-3l). These results indicate that the heavy atom effect of the anions has a negligible effect on the phosphorescence properties of cationic cellulose derivatives. In contrast, the strong hydrogen-bonding capability of the anions is a determining factor for the phosphorescence emission, because the strong hydrogen-bonding interactions inhibit the non-radiative transitions. These above results substantively and clearly confirm the phosphorescence mechanism. The imidazolium cations in Cell-ImCNCl promote the ISC process. The cyano-containing cations, chloride anions and hydroxyl groups interact with one another via hydrogen bonding interactions and electrostatic attraction interactions to inhibit the non-radiative transitions. As a result, the phosphorescence emission is achieved at room temperature.

Fig. R1 (a) Fluorophore spectra of CNMImCl (Ex = 365 nm). (b) RTP spectra of

CNMIImCl (Ex = 365 nm). (c) RTP lifetime spectra of CNMIImCl. (d) Photographs of CNMIImCl taken under 365 nm lamp and with the lamp off.

Fig. R2 (a) RTP spectra of CNMIImCl aqueous solution with a concentration of 1 mg/mL at different temperatures. (b) RTP lifetime spectra of CNMIImCl aqueous solution with a concentration of 1 mg/mL at different temperatures. (Ex = 365 nm)

Fig. R3 (a) RTP spectra of Cell-ImCNX (Ex = 320 nm). (b) Photoluminescence quantum yield of Cell-ImCNX with different anions. (c) RTP lifetime spectra of Cell-ImCNX with different anions.

For the real applications, excellent processing property and formability of the phosphorescence materials are essential. However, due to the rigid or complex structure, the previous reported phosphorescence materials generally exhibit poor processibility; therefore, they are difficult to satisfy various practical requirements. In our work, the new cationic cellulose derivative Cell-ImCNCl has excellent water solubility, therefore, it can be easily processed into various forms, including phosphorescent films, fibers, coatings, patterns and so on, by conventional processing methods, such as doctor blade coating, dip coating,

screen printing, inkjet printing and mask casting (Fig. 4). For instance, we can get the large area film (Fig. R4), which is much better and larger than the recent report (Large-Area, Flexible, Transparent, and Long-Lived Polymer-Based Phosphorescence Films, *J. Am. Chem. Soc.* **2021**, 143, 13675-13685). A large roll of phosphorescence fibers and numerous phosphorescence microspheres can be obtained by a dip coating method also (Fig. R5). In addition, the patterns can be formed on different substrates, including paper, ceramics, glass, plastic, stainless steel, and aluminum foil (Fig. 4). Such excellent processability and formability make phosphorescent Cell-ImCNCl applicable in many fields, such as complex anti-counterfeiting, information encryption and storage, smart labels, packaging, special fibers and detection sensors. These large-scale and various materials forms have not been reported in the previous works.

Fig. R4 The large-area phosphorescence film. (Ex = 365 nm)

Fig. R5 (a) The large roll of phosphorescence cellulose fibers. (b) Phosphorescence cellulose microspheres. (Ex = 365 nm)

In addition, for the real applications, excellent stability is crucial also. After adding a small amount of glutaraldehyde as the cross-linking agent, the as-fabricated phosphorescent materials exhibit excellent water resistance and

antibacterial properties (Fig. 5), which have not been achieved in the previous reports. In summary, cellulose-based easy-to-process RTP materials can act as antibacterial, water-resistant, and eco-friendly phosphorescent patterns, coatings and bulk materials, which have enormous potential in advanced anti-counterfeiting, information encryption, disposable smart labels, storage and monitoring of food and drugs, etc.

- (2) **Reviewer #1 wrote:** *For your system, cellulose itself needs to be used as a reference, those data have not been presented.*

Answer: Thanks for your kind suggestion. We have revised the section of experimental description and added the relevant experiment data in the revised manuscript as suggestion.

- (3) **Reviewer #1 wrote:** *For your system, what are the crystallinity changes when various of ionic structures introduced into the cellulose?*

Answer: Thanks for your kind suggestion. In order to observe the crystallinity changes after introducing various of ionic structures into the cellulose, we have obtained the XRD curves of the corresponding cellulose derivatives by a polycrystalline X-Ray diffractometer. The XRD curves of cellulose and Cell-ImCNCl are shown in Fig. R6. The cellulose has obvious diffraction peaks at 15.0° , 16.5° , 22.8° and 34.5° , corresponding to the (1-10), (110), (200) and (004) crystal planes of cellulose I crystal, respectively. When the ionic structures were introduced into the cellulose, the hydrogen bonding network in cellulose was broken, making it difficult for the obtained Cell-ImCNCl chains to arrange regularly. As a result, the obtained cellulose derivatives are amorphous, which is confirmed by the broad peak in the XRD curves (Fig. R6).

Fig. R6 (a) XRD curves of cellulose and Cell-ImCNCl with different DS_t and DS_{CN}; (b) XRD curves of cellulose and Cell-ImCNX.

(4) **Reviewer #1 wrote:** For your system, do you know the purity of the raw materials, does it affect the RTP?

Answer: Thanks for your kind suggestion. We know the previous report about the influence of trace impurities on the luminescence of carbazole. In our system, we washed our materials for more than four times by dissolving in DMSO and precipitating in acetone, and placed it in the vacuum oven for more than 48 hours at 60 °C. These processes ensure that there is no impurities and organic solvents in the samples, which can be proved in NMR spectra also.

(5) **Reviewer #1 wrote:** Try to cite some references on hydrogen bonds effects on RTP

Answer: Thanks for your kind suggestion. We have added some references on the hydrogen bonds effect on RTP as suggestion.

(6) **Reviewer #1 wrote:** The propose of using glutaraldehyde in the system as a crosslinking agent need to be further clarified, did you check the crystallinity, FTIR and others, also how about the purity of the system?

Answer: Thanks for your kind suggestion. The cross-linked Cell-ImCNCI is still amorphous (Fig. R7a). In the FTIR spectrum of glutaraldehyde (Fig. R7b), the peak at 2874 cm^{-1} is the characteristic absorption peak of C-H of aldehyde group, and the peak at 1712 cm^{-1} is the carbonyl stretching vibration peak. By comparing the FTIR spectra of glutaraldehyde and cross-linked Cell-ImCNCI, there is no relevant infrared characteristic peaks of glutaraldehyde. This can directly prove that there is no residual glutaraldehyde. Meanwhile, after cross-linking with glutaraldehyde, the hydroxyl stretching vibration peak is at 3354 cm^{-1} which shows a blue shift, and its intensity weakens, indicating the cross-linking occurs via the hydroxyl groups.

Fig. R7 (a) XRD curves of cellulose, uncross-linked Cell-ImCNCl and cross-linked Cell-ImCNCl; (b) FTIR spectra of glutaraldehyde, uncross-linked Cell-ImCNCl and cross-linked Cell-ImCNCl.

(7) **Reviewer #1 wrote:** *Antibacterial properties and the explanation is too brief, more evidence and reference needed to support your claim.*

Answer: Thanks for your kind suggestion. We have revised the antibacterial section as suggestion.

Answers to Comments by Reviewer # 2

Reviewer #2 wrote: *The manuscript is well written and describes very thoroughly the synthesis processes and the applications of an ultralong phosphorescence cellulose. The manuscript includes very interesting properties of the synthesized material, which corroborates the versatility of the applications at which the material can be used. However, even though the synthesis of the material is somewhat innovative, the new synthesis and the long phosphorescence exhibited by the cellulose based materials does not meet the novelty required for a publication in a journal such as Nature Communications. There are other journals specialized in materials where this manuscript would be a better suit. Just as examples, there are several research groups that have published similar materials in recent years, some of which have been cited by the authors in references 51-53. Therefore, my recommendation is to seek a more specialized journal to publish the work due to an insufficient novelty of the work to be published in Nature Communications.*

Answer: Indeed, because the organic room-temperature phosphorescence (RTP) has attracted much attention, there are many reports on organic RTP materials recently, and it has been shown that pure cellulose has a phosphorescence emission. But our work has many significant advance and advantages compared with the previous reports.

Firstly, we have proposed and demonstrated a new principle for the preparation of phosphorescent materials by combining the electrostatic attraction interactions with the hydrogen bonding interactions. The synergistical effect of the electrostatic attraction interactions with the hydrogen bonding interactions can effectively inhibit the non-radiative transitions, which is different to the previous methods, such as crystal structure, supramolecular assembling, encapsulating and cross-linking. Moreover, in our work, we found a new phosphor, the imidazolium

cation with the cyano group, which has not yet been reported in the previous work. This new system can provide a new direction for the design of new phosphors.

Secondly, this is a new type of phosphorescent material that was derived from natural biomass. Among the organic RTP materials reported, most of them are organic small molecules crystal, and only a small part of them is polymer materials. In the polymer-based phosphorescent materials, only PVA and PMMA were used as the substrates, while the biopolymer-based phosphorescent materials are extremely limited, because the strong hydrogen-bonding network in the biopolymers makes their modification and processing difficult. Although cellulose has been reported to have a phosphorescence emission (*Sci. China Chem.* 2013, 56, 1178-1182; *Macromol. Rapid Commun.* 2021, 42, 2100321), cellulose has obviously weak phosphorescence performance in fact. The average phosphorescence lifetime is 5 ms and the photoluminescence quantum yield is 3.76% (Fig. R8), because the hydrogen-bonding network is not strong enough and there is no phosphor in cellulose. Very recently, Tan et al. obtained organic amorphous polymers from alginate by introducing common organic phosphors (*Adv. Mater.* 2020, 2004768). Aida et al. prepared phosphorescent gelatin foam by freeze-drying method (*J. Am. Chem. Soc.* 2021, 143, 16256-16263), but the resultant phosphorescent materials are difficult to be processed. In our work, we enhance the hydrogen-bonding network meanwhile introduce the imidazolium cation phosphor to obtain cellulose-based phosphorescence materials with excellent processing property and formability by simply introducing a special ionic structure into cellulose chain.

Fig. R8 (a) RTP lifetime spectra of cellulose powder (Ex = 350 nm); (b) Photographs of cellulose powder taken under 365 nm lamp and with the lamp off.

In addition, the obtained phosphorescent Cell-ImCNCl can be easily processed into various forms, including phosphorescent films, fibers, coatings, patterns and so on, by conventional processing methods, such as doctor blade coating, dip coating, screen printing, inkjet printing and mask casting (Fig. 4). For instance, we can get the large area film (Fig. R4), which is much better and larger than the recent report (Large-Area, Flexible, Transparent, and Long-Lived Polymer-Based Phosphorescence Films, *J. Am. Chem. Soc.* **2021**, 143, 13675-13685). A large roll of phosphorescence fibers and numerous phosphorescence microspheres can be obtained by a dip coating method also (Fig. R5). In addition, the patterns can be formed on different substrates, including paper, ceramics, glass, plastic, stainless steel, and aluminum foil (Fig. 4). Such excellent processability and formability make phosphorescent Cell-ImCNCl applicable in many fields, such as complex anti-counterfeiting, information encryption and storage, smart labels, packaging, special fibers and detection sensors. These large-scale and various materials forms have not been reported in the previous works.

Furthermore, our phosphorescent material is composed of only a single component, which is different to the complex composite system in PVA-based

phosphorescence materials. Our phosphorescent material can completely avoid many problems of the composite system, such as inevitable migration and leaching, short lifetime, and poor stability.

Finally, it is a novel, simple, effective and general strategy to obtain processable phosphorescence materials from natural cellulose. The introduction of a special cation promotes the ISC effect and enhances the hydrogen-bonding work simultaneously, which achieves the effect of “1+1>2”. Based on this strategy, many new cellulose-based phosphorescence materials are found in our group. In addition, for the real applications, excellent stability is crucial also. After adding a small amount of glutaraldehyde as the cross-linking agent, our phosphorescent materials exhibit excellent water resistance and antibacterial properties (Fig. 5), which have not been achieved in the previous reports. We also prepared PVA/carbazole phosphorescence material according to the previous method (*J. Am. Chem. Soc.* 2021, 143, 13675-13685). But the obtained phosphorescence material is not water-resistant that restricts its use in daily life (Fig. R9). In summary, cellulose-based easy-to-process RTP materials can act as antibacterial, water-resistant, and eco-friendly phosphorescent patterns, coatings and bulk materials, which have enormous potential in advanced anti-counterfeiting, information encryption, disposable smart labels, storage and monitoring of food and drugs, etc.

Fig. R9 (a) Optical images of the PVA/carbazole phosphorescent pattern; (b)

Water-resistant test of the phosphorescent patterns.

Answers to Comments by Reviewer # 3

(1) **Reviewer #3 wrote:** *In this work, the authors reported cellulose-based easy-to-process RTP materials, which can be applied as antibacterial, water-resistant, and eco-friendly phosphorescent films, fibers, coatings, and patterns. The multiple hydrogen-bonding interactions and electrostatic attraction interactions of the RTP cellulose have efficiently suppressed the non-radiative transitions to achieve ultralong phosphorescence emission. Although **this work has some merits**, the novelty is rather limited and the data in the present are not convincing. Below are my concerns and suggestions.*

Answer: Many new experiments have been made to further demonstrate the novelty and reveal the phosphorescence mechanism. Our work has many significant advance and advantages compared with the previous reports as follows.

Firstly, we have proposed and demonstrated a new principle for the preparation of phosphorescent materials by combining the electrostatic attraction interactions with the hydrogen bonding interactions. The synergistical effect of the electrostatic attraction interactions with the hydrogen bonding interactions can effectively inhibit the non-radiative transitions, which is different to the previous methods, such as crystal structure, supramolecular assembling, encapsulating and cross-linking. Moreover, in our work, we found a new phosphor, the imidazolium cation with the cyano group, which has not yet been reported in the previous work. This new system can provide a new direction for the design of new phosphors.

Secondly, this is a new type of phosphorescent material that was derived from natural biomass. Among the organic RTP materials reported, most of them are organic small molecules crystal, and only a small part of them is polymer materials. In the polymer-based phosphorescent materials, only PVA and PMMA were used as the substrates, while the biopolymer-based phosphorescent materials are extremely limited, because the strong hydrogen-bonding network in the biopolymers makes their modification and processing difficult. Although cellulose has been reported to have a phosphorescence emission (*Sci. China Chem.* 2013, 56, 1178-1182; *Macromol. Rapid Commun.* 2021, 42, 2100321), cellulose has obviously weak phosphorescence performance in fact. The average phosphorescence lifetime is 5 ms and the photoluminescence quantum yield is 3.76% (Fig. R8), because the hydrogen-bonding network is not strong enough and there is no phosphor in cellulose. Very recently, Tan et al. obtained organic

amorphous polymers from alginate by introducing common organic phosphors (*Adv. Mater.* **2020**, 2004768). Aida et al. prepared phosphorescent gelatin foam by freeze-drying method (*J. Am. Chem. Soc.* **2021**, 143, 16256-16263), but the resultant phosphorescent materials are difficult to be processed. In our work, we enhance the hydrogen-bonding network meanwhile introduce the imidazolium cation phosphor to obtain cellulose-based phosphorescence materials with excellent processing property and formability by simply introducing a special ionic structure into cellulose chain.

In addition, the obtained phosphorescent Cell-ImCNCl can be easily processed into various forms, including phosphorescent films, fibers, coatings, patterns and so on, by conventional processing methods, such as doctor blade coating, dip coating, screen printing, inkjet printing and mask casting (Fig. 4). For instance, we can get the large area film (Fig. R4), which is much better and larger than the recent report (Large-Area, Flexible, Transparent, and Long-Lived Polymer-Based Phosphorescence Films, *J. Am. Chem. Soc.* **2021**, 143, 13675-13685). A large roll of phosphorescence fibers and numerous phosphorescence microspheres can be obtained by a dip coating method also (Fig. R5). In addition, the patterns can be formed on different substrates, including paper, ceramics, glass, plastic, stainless steel, and aluminum foil (Fig. 4). Such excellent processability and formability make phosphorescent Cell-ImCNCl applicable in many fields, such as complex anti-counterfeiting, information encryption and storage, smart labels, packaging, special fibers and detection sensors. These large-scale and various materials forms have not been reported in the previous works.

Furthermore, our phosphorescent material is composed of only a single component, which is different to the complex composite system in PVA-based phosphorescence materials. Our phosphorescent material can completely avoid many problems of the composite system, such as inevitable migration and leaching, short lifetime, and poor stability.

Finally, it is a novel, simple, effective and general strategy to obtain processable phosphorescence materials from natural cellulose. The introduction of a special cation promotes the ISC effect and enhances the hydrogen-bonding work simultaneously, which achieves the effect of “1+1>2”. Based on this strategy, many new cellulose-based phosphorescence materials are found in our group. In addition, for the real applications, excellent stability is crucial also. After adding a

small amount of glutaraldehyde as the cross-linking agent, our phosphorescent materials exhibit excellent water resistance and antibacterial properties (Fig. 5), which have not been achieved in the previous reports. We also prepared PVA/carbazole phosphorescence material according to the previous method (*J. Am. Chem. Soc.* 2021, 143, 13675-13685). But the obtained phosphorescence material is not water-resistant that restricts its use in daily life (Fig. R9). In summary, cellulose-based easy-to-process RTP materials can act as antibacterial, water-resistant, and eco-friendly phosphorescent patterns, coatings and bulk materials, which have enormous potential in advanced anti-counterfeiting, information encryption, disposable smart labels, storage and monitoring of food and drugs, etc.

In order to understand the mechanism clearly, the 1-cyanomethylimidazolium chloride (CNMImCl) was synthesized and characterized firstly. The CNMImCl solid powder exhibits a similar phosphorescence emission to that of Cell-ImCNCl (Fig. R1). Moreover, the CNMImCl aqueous solution (1 mg/mL) has a phosphorescence emission at 77-137 K, while gives a phosphorescence quenching at 197-273 K (Fig. R2). These phenomena confirm that the imidazolium cation with the cyano group and nitrogen element promotes the intersystem crossing (ISC). Further, we replaced 1-cyanomethylimidazole with 1-butylimidazole to prepare a new cationic cellulose derivative, cellulose 1-butylimidazolium chloride (Cell-BimCl) (Fig. S6). Cell-BimCl has very poor phosphorescence performance, and its average phosphorescence lifetime is only 5 ms, which is much lower than the average phosphorescence lifetime of Cell-ImCNCl (158 ms). Compared with the cyanomethyl group, the butyl group in the butylimidazolium cation is a long alkyl segment with a lack of π -conjugated system, lone-pair electron and hydrogen bond forming ability, which weakens both ISC process and hydrogen bonding interactions. Due to the lack of strong hydrogen bonding interactions to fix the long butyl segment, the triplet energy can be easily dissipated via the rotation and vibration of the butyl segment (Fig. S7). Therefore, Cell-BimCl has a poor phosphorescence emitting performance. In addition, for 1-(cyanomethyl)imidazole, since there is no confinement effect originating from the polymer chains and strong interactions, the non-radiative transitions occur. Therefore, 1-(cyanomethyl)imidazole has no phosphorescence (Fig. S8).

Furthermore, the cellulose 1-cyanomethylimidazolium fluoride

(Cell-ImCNF) with negligible heavy atom effect and strong hydrogen-bonding basicity of fluoride ion exhibits a similar phosphorescence quantum yield and phosphorescence lifetime to Cell-ImCNCl (Fig. R3), while the cellulose 1-cyanomethylimidazolium bromide (Cell-ImCNBr) with strong heavy atom effect and weak hydrogen-bonding basicity of bromide ion gives a significantly decreased phosphorescence quantum yield and lifetime (Fig. 3k and 3l). If the Cl⁻ anion was replaced with the anions with weaker hydrogen bond acceptor abilities, such as BF₄⁻, PF₆⁻, Tf₂N⁻, C(CN)₃⁻, N(CN)₂⁻, SCN⁻ (Fig. 3i). As a result, the resultant new cationic cellulose derivatives Cell-ImCNX have reduced phosphorescence performance compared with Cell-ImCNCl (Fig. 3j-3l). These results indicate that the heavy atom effect of the anions has a negligible effect on the phosphorescence properties of cationic cellulose derivatives. In contrast, the strong hydrogen-bonding capability of the anions is a determining factor for the phosphorescence emission, because the strong hydrogen-bonding interactions inhibit the non-radiative transitions. These above results substantively and clearly confirm the phosphorescence mechanism. The imidazolium cations in Cell-ImCNCl promote the ISC process. The cyano-containing cations, chloride anions and hydroxyl groups interact with one another via hydrogen bonding interactions and electrostatic attraction interactions to inhibit the non-radiative transitions. As a result, the phosphorescence emission is achieved at room temperature.

(2) Reviewer # 3 wrote: *To achieve phosphorescence emission, the authors chose natural cellulose as the raw material and introduced cyanomethylimidazolium cations (ImCN⁺) and chloride anions (Cl⁻) into the cellulose chain to obtain the RTP materials. However, cellulose for sustainable phosphorescent materials has already been reported (Carbon 171 (2021) 946-952; ACS Nano 2020, 14, 11130-11139). Therefore, the novelty of this work is limited.*

Answer: We have carefully studied these two articles. These two articles are instructive for the development of cellulose-based phosphorescent materials. However, in these two articles, carbon dots are phosphor while cellulose nanocrystals are the substrate only. Moreover, the obtained hybrid materials have not exhibited a good formability, as shown in these two articles. Our phosphorescent material is composed of only a single component, which is different to these complex composite systems. Our phosphorescent material can completely avoid many problems of the composite systems, such as inevitable

migration, poor homogeneity, and short lifetime. In addition, the obtained phosphorescent Cell-ImCNCl can be easily processed into various forms, including phosphorescent films, fibers, coatings, patterns and so on, by conventional processing methods, such as doctor blade coating, dip coating, screen printing, inkjet printing and mask casting (Fig. 4). For instance, we can get the large area film (Fig. R4), which is much better and larger than the results in these two articles (*Carbon* 171 (2021) 946-952; *ACS Nano* 2020, 14, 11130-11139). A large roll of phosphorescence fibers and numerous phosphorescence microspheres can be obtained by a dip coating method also (Fig. R5). In addition, the patterns can be formed on different substrates, including paper, ceramics, glass, plastic, stainless steel, and aluminum foil (Fig. 4). Such excellent processability and formability make phosphorescent Cell-ImCNCl applicable in many fields, such as complex anti-counterfeiting, information encryption and storage, smart labels, packaging, special fibers and detection sensors. These large-scale and various materials forms have not been reported in the previous works.

More importantly, our work demonstrates a new principle for the preparation of phosphorescent materials. It is a novel, simple, effective and general strategy to obtain processable phosphorescence materials from natural cellulose. The introduction of a special cation promotes the ISC effect and enhances the hydrogen-bonding work simultaneously, which achieves the effect of “1+1>2”. Based on this strategy, many new cellulose-based phosphorescence materials are found in our group. Therefore, we believe that our materials are innovative and have broad application prospects in the future.

- (3) **Reviewer # 3 wrote:** *In page 7, the author increased degree of substitution of imidazolium cation (DS_{CN}). It causes efficiency and lifetime of phosphorescence enhanced first and then decreased. However, when the DS_{CN} is 0.37, the corresponding efficiency and lifetime contradict the laws proposed by author. Why could the unique phenomenon be observed in 0.37?*

Answer: Thanks for your kind suggestion. In the overall DS range of 0.31-1.0, it is obvious that the efficiency and lifetime of phosphorescence enhanced first and then decreased (Fig. 3b-3d). When the DS is in a range of 0.31-0.42, because the change of DS value is only 0.11, there are small changes in the phosphorescence efficiency and lifetime. The slight difference between these data may be caused

by the error of the instrument measurement, and it does not affect the discussion of the overall change rule.

- (4) **Reviewer # 3 wrote:** *In page 8. The author kept DS_{CN} and changed DSt , but the experiment results of Fig. 3h lacks of regularity. Therefore, the author proposed the most appropriate value of 1.24 for DSt is probably not accurate.*

Answer: Thanks for your kind suggestion. We have revised the description in the section of results and discussion as suggestion.

- (5) **Reviewer # 3 wrote:** *The authors made a "plane" with Cell-ImCNCl/glutaraldehyde on a glass substrate, after being immersed into water for 24 hours; the "plane" pattern had a negligible change. The authors demonstrated that the Cell-ImCNCl has high water-resistant property. However, whether the material changes after immersed into water requires more microscopic observation such as observation by microscope or scanning electron microscope. Besides, more rigorous testing methods such as the water contact angle of Cell-ImCNCl also should be carried out to prove water-resistant.*

Answer: Thanks for your kind suggestion. In order to confirm the water-resistant property, the SEM images of the crosslinked Cell-ImCNCl film before and after immersion in water have been shown in Fig. R10. The cross-linked Cell-ImCNCl film has the dense structure without defects on the surface. After immersion in water, there is no change on the surface (Fig. R10).

The contact angle photos of cross-linked Cell-ImCNCl for different times are shown in Fig. R10. As the water drops on the surface of Cell-ImCNCl, the initial water contact angle is 60° , becomes 39.4° after 0.5 hour and 36.8° in the end. Due to the cross-linking effect of glutaraldehyde, our material has good water resistance. Although as the time goes by, the contact angle becomes smaller because the water droplet interacts with more hydroxyl groups in the material, eventually, the water droplet does not disappear and the contact angle maintains at a constant angle. Furthermore, the optical photographs of cross-linked Cell-ImCNCl with dropped water for different times are obtained (Fig. R10). The water droplet exists on the surface of the film as the time prolongs, further proving the water resistance of our material. Thus, these above experimental results prove that our material has good water resistance.

Fig. R10 (a-c) SEM images of cross-linked Cell-ImCNCl at different magnifications before immersed into water; (d-f) SEM images of cross-linked Cell-ImCNCl at different magnifications after immersed into water; (g-i) Contact angle photos of cross-linked Cell-ImCNCl at different times: (g) 0 h, (h) 0.5 h, (i) 2 h; (j and k) Optical photographs of cross-linked Cell-ImCNCl with water droplet at different times: (j) 0 h, (k) 2 h.

(6) **Reviewer # 3 wrote:** *The authors detected the antibacterial properties of phosphorescent pattern by LB agar colony in Figure 5d and elucidated that the excellent antibacterial ability probably originated from a high ion concentration environment around the phosphorescent pattern. While, the phosphorescent pattern is composed of Cell-ImCNCl/glutaraldehyde aqueous solution, it is well known that glutaraldehyde is commonly used for biological fixation and is highly lethal to bacteria. Therefore, the authors should rule out the effect of glutaraldehyde on antibacterial activity.*

Answer: Thanks for your kind suggestion. In our system, we use a small amount of glutaraldehyde as the cross-linking agent. After cross-linking, the unreacted glutaraldehyde is completely removed by vacuum heating and drying with enough time. In order to check whether there is glutaraldehyde, the FTIR spectra of uncross-linked and cross-linked Cell-ImCNCI were measured (Fig. R11). The C-H characteristic peak of aldehyde group is at 2874 cm^{-1} , and the carbonyl peak is at 1712 cm^{-1} . By comparing the FTIR spectra of glutaraldehyde and cross-linked Cell-ImCNCI, there is no relevant characteristic peaks of glutaraldehyde can be found. This can directly prove that there is no residual glutaraldehyde in the cross-linked sample.

Fig. R11 FTIR spectra of glutaraldehyde, uncross-linked Cell-ImCNCI and cross-linked Cell-ImCNCI.

(7) **Reviewer # 3 wrote:** *The authors clarify that the obtained organic RTP material has much excellent properties such as easy-to-process, antibacterial, water-resistant, and environmentally-friendly. What role do these features play in the specific application, or how do they relate to each other? It should be further elaborated.*

Answer: Thanks for your kind suggestion. At different application fields, different properties are needed. It is not necessary to have all of these properties at the same time everywhere. For example, when we prepare membranes and

fibers at a large scale, we need materials that are easy to process, soluble, and environmentally-friendly. For another example, when the phosphorescent materials are used as labels and coatings, the excellent processability and environmental-friendliness are of significant importance, because the usage quantity of phosphorescent materials as labels and coatings is too small to recycle or reuse. In addition, the anti-bacterial property makes the phosphorescent materials be used as a food anti-counterfeiting label. As a coating, the anti-bacterial property can avoid a contamination from bacteria. The materials from nature can be disposable due to their environmental friendliness. These excellent properties of our material can satisfy the different applications in many fields.

- (8) **Reviewer # 3 wrote:** *The abbreviation of a noun should be given its full name when it first appears. Such as, Page 12, lines 266 and 267, Escherichia coli (E. coli) and Staphylococcus aureus (S. aureus). The name of bacteria should be in italics.*

Answer: Thanks for your kind suggestion. We have revised them in the manuscript as suggestion.

- (9) **Reviewer # 3 wrote:** *The description of the figure annotations should be more detailed and easier for readers to understand. For example, Fig. 5d, the antibacterial assay method should be included.*

Answer: Thanks for your kind suggestion. We have revised them in the manuscript as suggestion.

REVIEWERS' COMMENTS

Reviewer #1 (Remarks to the Author):

I am happy with the authors' response to my comments.

[Editorial Note: The reviewer was asked to look also over the comments given in response to reviewer #3, who was not available to look on the revision again.]

Answers to Comments by Reviewers

Answers to Comments by Reviewer # 1

(1) **Reviewer #1 wrote:** *I am happy with the authors' response to my comments.*

Answer: Thanks for your nice comments.

Answers to the Format Requirement

- (1) The Reporting Summary was revised as requirement.
- (2) The editable figures have been provided in a PowerPoint file.
- (3) The Supplementary Information was provided as a PDF file.
- (4) The Abstract Section was shortened to 150 words in the revised manuscript.
- (5) The Methods Section was revised as requirement.
- (6) The ORCID of Jinming Zhang is 0000-0003-3404-4506; the ORCID of Jun Zhang is 0000-0003-4824-092X.